

# A method of water resources accounting based on deep clustering and attention mechanism under the background of integration of public health data and environmental economy

Shiya Zhou

Wuhan Technology and Business University, Wuhan, Hubei, China

## ABSTRACT

Water resource accounting constitutes a fundamental approach for implementing sophisticated management of basin water resources. The quality of water plays a pivotal role in determining the liabilities associated with these resources. Evaluating the quality of water facilitates the computation of water resource liabilities during the accounting process. Traditional accounting methods rely on manual sorting and data analysis, which necessitate significant human effort. In order to address this issue, we leverage the remarkable feature extraction capabilities of convolutional operations to construct neural networks. Moreover, we introduce the self-attention mechanism module to propose an unsupervised deep clustering method. This method offers assistance in accounting tasks by automatically classifying the debt levels of water resources in distinct regions, thereby facilitating comprehensive water resource accounting. The methodology presented in this article underwent verification using three datasets: the United States Postal Service (USPS), Heterogeneity Human Activity Recognition (HHAR), and Association for Computing Machinery (ACM). The evaluation of Accuracy rate (ACC), Normalized Mutual Information (NMI), and Adjusted Rand Index (ARI) metrics yielded favorable results, surpassing those of K-means clustering, hierarchical clustering, and Density-based constraint extension (DCE). Specifically, the mean values of the evaluation metrics across the three datasets were 0.8474, 0.7582, and 0.7295, respectively.

# INTRODUCTION

Water resources accounting is the statistical method of water resources in a certain basin to carry out statistics of various indicators, which is of great significance for water resources management and environmental protection (*Li & Qian, 2018*). With the rapid development of industry and economy, water pollution has become an increasingly important social problem (*Zamora-Ledezma et al., 2021*). Nowadays, people pay more attention to integrating public health and safety as well as the environmental economy. In the accounting of water resources, water resources liabilities are the indicators that can best

Corresponding author
Shiya Zhou,
syzhouzhou9090@163.com

reflect the public health of water resources and the environmental economy (*Zhang, Shen & Sun, 2022*). The needs of public safety and economic environment make the upgrading and optimization of water resources management system a hot topic for discussion.

Traditional accounting relies on human resources to analyze and classify some indicators, and the workload of this method is undoubtedly huge (*Ermakova, Oznobihina & Avilova, 2020*). Manual accounting not only needs to process a large amount of data, but also requires the absolute seriousness of the staff, and a little mistake can lead to a large number of deviations in the overall data. Computer technology can quickly complete a large number of calculations, and at the same time has the advantage of high accuracy. But the traditional computer lacks the automatic algorithm to carry on the necessary calculation to the key part of water resources accounting. How to use modern science and technology to assist accounting in the context of public health and safety and environmental and economic integration, so as to make it more intelligent? Automation is the problem we need to solve (*Lacity & Willcocks, 2021*).

With the development of the computer industry, machine learning (*Sammut & Webb, 2010*) has been widely used in various fields. Machine learning is a multidisciplinary discipline, including design probability theory, statistics, approximation theory, algorithm complexity theory, *etc*. Its research center is to enable computer simulation to realize human behavior, and has made great contributions to automation in many fields. Clustering algorithm in machine learning provides convenience for data analysis in many fields. These methods include K-means algorithm (*Wang & Su, 2011*), hierarchical clustering (*Murtagh & Contreras, 2017*), density-based spatial clustering of applications with noise (DBSCN) (*Bäcklund & Hedblom, 2011*), mean shift clustering (*Anand & Mittal, 2013*), *etc. Ying & Chong (2010)* used mean shift algorithm to realize unsupervised segmentation of aperture radar data. *Bouguettaya & Yu (2015)* used hierarchical clustering method to group a large number of sensors unsupervised, so that each root transmitter can communicate with the cluster head. *Sinaga & Yang (2020)* used the improved K-means to implement a clustering algorithm that does not require user parameter description and is applied to the recognition system of isolated words. Although these clustering methods based on machine learning can provide convenience in the field of accounting, most features in machine learning need expert recognition and manual coding.

Deep learning (*LeCun & Bengio, 2015*) has gradually become a popular application in various fields with its end-to-end problem solving method. It uses the powerful feature extraction ability of convolution operation to build a convolutional neural network (*Li et al., 2021*), which is applied to learning modules in various fields, including clustering. Deep Embedded Clustering (DEC) model (*Ozanich & Thode, 2021*) uses the fixed length power spectrum of signals to learn potential features and form classification clustering, which is widely used in various data analysis fields. The Improved Deep Embedded Clustering (IDEC) model (*Guo et al., 2017*) adds a reconstruction loss to the DEC model to learn better representation. *Ajay et al. (2022)* proposed a deep clustering framework based on pairwise data similarity, its performance exceeds that of some existing methods of the same type. However, the receptive field of simple convolution operation is limited, and

there is a lack of global information for the classification of water resources liabilities requiring complete feature classification.

To solve the above problems, we propose an auxiliary method of water resources accounting based on deep clustering and attention mechanism (*Ajay et al., 2022*). This method is mainly aimed at the key part of water resource liability accounting in the context of public health data and environmental economic integration. In this article, the index of water resource liability is constructed by feature, and then a deep clustering algorithm is constructed by convolutional operation, and an improved Non-Local module is inserted. Proposed method can automatically cluster the debt levels of water resources, thus facilitating the subsequent debt analysis. This automatic method can greatly save the manpower in the classification of liabilities, and at the same time has high accuracy. Therefore, the proposed method is verified on several common datasets containing complex data, and compared with some traditional and deep learning clustering methods. Experiments show that our method has better performance and has certain application value in the field of accounting.

# RELATED WORK

## Deep clustering methods

Deep clustering method is an end-to-end unsupervised clustering method based on deep learning technology. Compared with the supervised deep learning method, unsupervised learning is suitable for applications that lack sufficient prior knowledge. Besides, it is difficult to label categories of the supervised deep learning method manually (*Guo et al., 2022*). Water resources accounting data is large and complex, and it is difficult to obtain high quality labels through labeling. In similar difficult to label work, predecessors have achieved unsupervised problem solving through different deep clustering methods. *Denny & Spirling (2018)* proves that clustering algorithm can reduce the dependence on annotations and turn the method that needs strong supervision algorithm implementation into weak supervision implementation. Instead of turning strong supervision into weak supervision, *Iakovidis & Georgakopoulos (2018)* analyzed data-driven phenotypes of patients with heart failure using simple deep clustering method, echocardiography and electronic health record data. However, the volume of health data is small in other areas, *Ulloa & Wehner (2017)* used the deep clustering method to visually supervise the representation learning of large unlabeled data sets, and realized short-term traffic prediction and applied it to the intelligent transportation system. These deep learning clustering methods can automatically and intelligently complete some data analysis work. However, compared with these work, in the water resources accounting work under the background of public health data and environment integration, the feature matrix composed of data is more integrated, and more global information is needed to increase the accuracy of clustering, so as to more accurately grade the water resources liabilities in various regions.

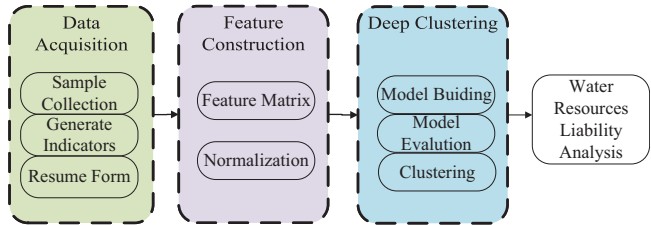

**Figure 1 Overall flow of water resource liability analysis with deep clustering.**

## Attention mechanism

An important significance of attention mechanism in the field of vision is to give global information to the feature map, and the receptive field of convolution operation is expanded to the whole map. In the field of accounting, the feature matrix generated by the original data has the same organizational structure as the image in the visual field, which also requires the addition of global information. *Han & Zheng (2019)* built an end-to-end model using self-attention mechanism and convolutional neural network to solve machine reading and question answering problems. The attention mechanism method used in the model is integrated into convolution operation, so as to focus more on difficult samples to improve the performance of the model. *Wei & Dohan (2018)* proposed a self-attention generation countermeasure network that allows attention driven remote dependency modeling of image generation tasks. This is consistent with our concept of adding global information to the network. *Zhang & Goodfellow (2019)* proposed a non-local module, which can be directly inserted into the convolutional neural network to collect rich global information for the network. As a separate module, Non-local can well improve the performance of the model and is widely used in supervised methods. However, this module has the problems of large computation and redundant information. These problems make it difficult for non-local to act on larger characteristic matrices. The water resources liability characteristic matrix has a large scale. If you want to use non-local to fuse the global information, you need to improve its structure (*Wang & Girshick, 2018*).

## METHODS

To automate and intelligently conduct water resource accounting within the context of integrating public health data and environmental economy, we propose a methodology aimed at aiding in the assessment of water resource liabilities. The schematic representation of the overall methodology can be observed in Fig. 1. Initially, water samples are collected, and several water quality evaluation indicators, which directly impact water resource liabilities, are generated. Subsequently, water quality indicator tables are constructed. The subsequent stage involves the creation of features based on the generated table, resulting in the generation of a feature matrix, which is subsequently normalized. The third phase entails the development of a deep clustering model, followed by model evaluation and unsupervised clustering of the constructed features. Ultimately, the outcomes of deep clustering are utilized to evaluate the debt level of water resources across various regions. Based on the debt level of water resources, distinct accounting

**Table 1 Raw form of water quality.**

| Sample number | Chl. a | pH | SD | DO | TN |
|---|---|---|---|---|---|
| S1 | C1 | P1 | SD1 | D1 | T1 |
| S2 | C2 | P2 | SD2 | D2 | T2 |
| S3 | C3 | P3 | SD3 | D3 | T3 |
| S4 | C4 | P4 | SD4 | D4 | T4 |
| S5 | C5 | P5 | SD5 | D5 | T5 |

Note:
Table 1 shows the sample data format of multiple samples in a certain region. The input format of the neural network is in the form of eigenvectors.

indicators are employed to conduct water resource liability accounting within the framework of integrating public health data and environmental factors, thereby facilitating comprehensive accounting procedures.

## Feature structure

We selected several water quality indicators that have a greater impact on water resource liabilities as the evaluation criteria for the classification of water resource liabilities: chlorophyll (Chl. a), pH value, transparency (SD), dissolved oxygen (DO), total oxygen (TN) and total phosphorus (TP). After collecting water samples from a certain area and detecting the original data, it is necessary to establish the original data table. Table 1 shows the sample data format of multiple samples in a certain region. Each column of the table represents the data of a different sample, and each row represents a different water quality index of a sample. The input format of the neural network is in the form of eigenvectors. In order to adapt the original data to the needs of the deep clustering model input, we use the data in the table to generate a feature matrix, as shown in Formula (1). The matrix encompasses a sequential incorporation of data into the sample data table, with each characteristic matrix representing the water quality attributes specific to a given geographical region. It is noteworthy that the number of samples surpasses the mere five exemplified in the formula when a sufficient quantity of samples is procured. In order to facilitate gradient calculation within the neural network framework, the feature matrix undergoes a normalization process. The normalization of the feature matrix serves to eradicate the potential influence arising from variations in units and scales among the distinct features, thereby ensuring equitable treatment of each dimension's feature. Consequently, this normalization procedure manifests an enhanced convergence velocity and ultimate accuracy for the model.

The complete feature building process is shown in Fig. 2.

$$\begin{pmatrix} C1 & P1 & SD1 & D1 & T1 \\ C2 & P2 & SD2 & D2 & T2 \\ C3 & P3 & SD3 & D3 & T3 \\ C4 & P4 & SD4 & D4 & T4 \\ C5 & P5 & SD5 & D5 & T5 \end{pmatrix} \tag{1}$$

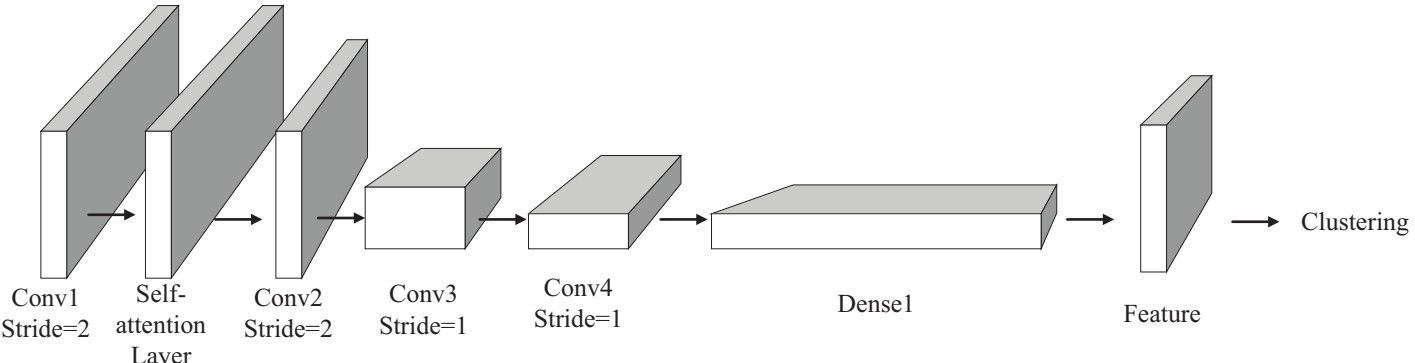

| Data Acquisition | → | Feature Table | → | Feature Matrix | Normalization | Input Feature |

**Figure 2 Schematic diagram of feature construction process.**

Conv1
Stride=2

Self-attention Layer

Conv2
Stride=2

Conv3
Stride=1

Conv4
Stride=1

Dense1

Feature

Clustering

**Figure 3 Structure diagram of deep clustering model.**

## Clustering model of deep learning with attention mechanism

In order to improve the accuracy of clustering and reasonably classify the calculation level of water resources liabilities, we propose a deep clustering model with attention mechanism. The model learns by self-monitoring, extracts the features of the original data with convolution operation, and classifies the water resource liability level of a region by integrating these features. The overall structure of the deep clustering model is shown in Fig. 3. The model has four convolution layers for feature extraction.

The self-attention mechanism module, which we have incorporated, is positioned between the first and third convolutional layers. This module serves the purpose of incorporating global information into the feature extraction process. A comprehensive explanation of the Self-Attention module will be provided in the subsequent section. Following the completion of the feature extraction process, the model expands the extracted features into a one-dimensional long vector, with the second dimension set to 1. This vector then enters the fully connected layer to obtain a low-dimensional representation of the features. Subsequently, the acquired features are fed into the final clustering layer, where the clustering algorithm carries out the ultimate classification. Diverging from supervised deep learning models, our unsupervised deep clustering algorithm employs the reconstruction error function as the loss function, as depicted in Eq. (2).

$$L = \frac{1}{n}\sum_{i=1}^{n}||x_i - \tilde{x}_i||^2 + \lambda \sum ||\theta||_2^2 \tag{2}$$

At the last clustering layer, we use the classical K-means algorithm to perform the final clustering. First, we use the soft assignment based on $t$ distribution to calculate the similarity $s_{ij}$ between the position feature representation $z_i$ and the cluster center point $u_j$. The calculation formula is $s_{ij} = \left(1 + \frac{w_j \times d_{ij}}{\alpha}\right)^{-\frac{a+1}{2}}$, where $d_{ij} = ||z_i - u_i||_2^2$ is the Euclidean distance between the feature representation $z_j$ and the cluster center point $u_j$, and $\alpha > 0$ is the degree of freedom of the $t$ distribution; In addition, the super parameter $w_j = \begin{cases} 1, d_{ij} = \min\{d_{i1}, d_{i2}, ..., d_{ik}\} \\ 2, others \end{cases}$ increases the similarity between the nearest feature $z_i$ and the cluster center $u_j$. The probability value obtained after $s_{ij}$ normalization is used as the soft assignment of clustering, as shown in Formula (3):

$$q_{ij} = \frac{s_{ij}}{\sum\limits_{k=1}^{K} s_{ik}} \tag{3}$$

Secondly, auxiliary distribution $q_{ij}$ is constructed from soft distribution $p_{ij}$ to make auxiliary distribution $p_{ij}$ more reliable, as shown in Formula (4):

$$p_{ij} = \frac{q_{ij}^{\beta}}{\sum\limits_{k=1}^{K} q_{ik}^{\beta}} \tag{4}$$

where $\beta$ is an exponential power. Two probability distributions $p_i$ and $q_i$ are obtained from Eqs. (3) and (4). When the probability value $q_i$ of a class cluster in $q_{ic}$ is greater than $1/K$, the probability value of the corresponding $p_{ic}$ must be greater than $q_{ic}$, and the corresponding probability of other positions of $p_i$ is less than $q_i$. This shows that auxiliary distribution $p_i$ improves the probability of belonging to a certain cluster, while reducing the probability of belonging to other clusters, and improves the reliability of clustering.

The method described in this section mainly uses convolution operations for feature extraction, and then unsupervised clustering of the extracted representations. In addition, the traditional K-means clustering is improved to make it have higher confidence. This model can automatically grade water debt data.

## Improved non-local self-attention module

Although convolution operation has strong feature extraction ability, its receptive field is limited. In order to add global information to the feature extraction process, that is, to associate the elements in the feature matrix with all other elements, we introduce the self-attention module. The original non-local module is plagued by large number of parameters and difficulty in calculation. In order to solve this problem, we have made some improvements to the original structure to improve the operation efficiency. The overall structure of the self-attention module is shown in Fig. 4.

The specific process consists of the following steps:

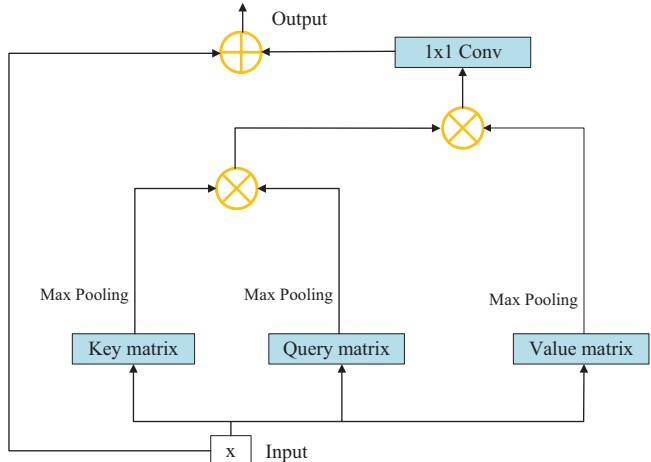

**Figure 4** **Structure diagram of self-attention module.**

Step 1: Initially, the input characteristic matrix undergoes a convolution operation to derive key, query, and value matrices. Each of these matrices captures distinct aspects of the input data.

Step 2: Subsequently, the maximum average pooling method is employed to sample the aforementioned matrices, effectively eliminating redundant information and reducing the parameter count. This technique enhances the feature extraction capability of the model.

Step 3: The key and query matrices are pointwise multiplied to yield a weight matrix. This matrix represents the importance or relevance of different elements within the input data.

Step 4: The weight matrix is then normalized using the softmax function, resulting in weight coefficients. This normalization step ensures that the coefficients sum up to 1 and facilitates proper weighting of the different elements.

Step 5: The weight coefficients are multiplied element-wise with the value matrix, and channel attention is applied to the corresponding positions across all channels of each characteristic matrix. This step allows for refining and highlighting relevant features within the input data.

Step 6: Finally, the resulting characteristic matrix, which embodies the refined and attended features, is outputted as the final representation of the data. Among them, the amount of calculation for sampling key, query and value matrices is greatly reduced. Because redundant information affecting accuracy is removed at the same time, experiments show that this operation makes the module have a certain amount of improvement in performance.

## EXPERIMENTS AND ANALYSIS

In order to evaluate the clustering effect of the proposed method, we use some common data sets to evaluate (https://zenodo.org/record/7924095#.ZFyzJHZByNc): (1) The USPS

data set contains 9,298 tables 16 × 16 pixel gray handwritten digital image These features are the gray values of pixels in the image, and all features are normalized to [0,1]. (2) The Heterogeneous Human Activity Recognition (HHAR) dataset contains 10,299 sensor records from smartphones and smartwatches. All samples were divided into six categories of human activities, including riding, sitting, standing, walking, going up and down stairs. (3) ACM is from the article network of ACM dataset. If two articles are written by the same author, there is an edge between them. The article features a bag of keywords. We selected the articles published in KDD, SIGMOD, SIGCOM and MobiCOMM and divided them into three categories (database, wireless communication and data mining) according to their research fields. All experiments were completed in the following environments: Intel (R) Xeon (R) Bronze 3204 CPU @ 1.90 GHz, 32 GB RAM, GPU Tesla V100, CentOS Linux release 7.6.1810.

In the experiment, Adam algorithm is used for optimization, Dynamic Decay Learning Rate and Batch Normalization are used.

## Evaluation metrics

For clustering results, we use common centralized evaluation indicators to measure the performance of the algorithm: accuracy rate (ACC), regular mutual information (NMI), and RAND index (ARI).

(1) The accuracy rate (ACC) is used to represent the correctly classified samples.

$$ACC = \frac{\sum_{i-1}^{N} \delta(map(l_i) = y_i)}{N} \tag{5}$$

where $\delta$ is an indicator function; $l_i$ is the clustering index of $x_i$ obtained by the algorithm; $y_i$ is the true label of $x_i$; $map$ is a transformation function, which maps each cluster label $l_i$ to a category as a whole, making the predicted label closer to the real label as a whole. This process can be obtained through Kuhn Munkres.

(2) Normalized mutual information (NMI) is another commonly used measure of clustering effect, which is used to represent the regularization amount of real label information that can be obtained after knowing the predicted clustering results.

$$NMI(Y, L) = \frac{I(Y, L)}{\sqrt{H(Y)H(L)}} \tag{6}$$

where $Y$ and $L$ correspond to real tags and predictive tags. $H$ is the information entropy function and $\sqrt{H(Y)H(L)}$ is the regularization $l$ term.

(3) The adjusted rand index (ARI) is used to measure the coincidence of two data distributions.

$$ARI = \frac{RI - E[RI]}{\max(RI) - E(RI)} \tag{7}$$

$$RI = \frac{a + b}{C_n^2} \tag{8}$$

**Table 2 Comparison of clustering performance between our method and existing methods.**

| Dataset | Metric | K-means (*Zamora-Ledezma et al., 2021*) | Hierarchical clustering (*Zhang, Shen & Sun, 2022*) | DCE (*Anand & Mittal, 2013*) | Ours |
|---------|--------|-------------|------------------------|--------|------|
| USPS | ACC | 0.6682 | 0.6753 | 0.7311 | 0.7841 |
|      | NMI | 0.6263 | 0.6456 | 0.7058 | 0.7898 |
|      | ARI | 0.5455 | 0.5622 | 0.6378 | 0.7065 |
| HHAR | ACC | 0.5988 | 0.6012 | 0.6978 | 0.8675 |
|      | NMI | 0.5845 | 0.5932 | 0.7351 | 0.8176 |
|      | ARI | 0.4578 | 0.4453 | 0.6089 | 0.7654 |
| ACM  | ACC | 0.6742 | 0.6753 | 0.8563 | 0.8906 |
|      | NMI | 0.3278 | 0.3521 | 0.5643 | 0.6673 |
|      | ARI | 0.3087 | 0.3298 | 0.6162 | 0.7167 |

**Note:**
Which depends on the feature extraction ability of convolutional neural network and the global information collection of attention mechanism. Compared with the DEC method of deep learning clustering method, the proposed method has also achieved better performance in the three data sets.

where, $n$ represents the total number of samples, $C$ represents the correct classification, and $K$ represents the predictive clustering results. $a$ is defined as the number of instance pairs that are divided into the same category in $C$ and the same cluster in $K$. $b$ is defined as the number of instance pairs divided into different categories in $C$ and different clusters in $K$. The value range of ARI is $[-1,1]$. The larger the value, the better the clustering effect.

## Comparison of different methods

We choose several existing clustering methods to compare with our proposed methods: (1) K-means, a classical clustering method based on original data. (2) Hierarchical clustering, tree clustering structure, classical algorithm. (3) DEC, a deep clustering method, designed a clustering target to guide the learning of data representation. See Table 2 for the evaluation indicators and comparison of several methods. It can be seen that our method greatly surpasses the two traditional clustering methods in performance, which depends on the feature extraction ability of convolutional neural network and the global information collection of attention mechanism. Compared with the DEC method of deep learning clustering method, the proposed method has also achieved better performance in the three data sets. The experimental results confirm the effectiveness and practical value of the proposed method. The proposed method can make considerable contributions to the accounting of water resources under the background of public data security and environmental economy integration.

## Clustering visual analysis

In order to more intuitively reflect the performance advantages of our proposed methods, we have made visual analysis on different data sets for the four methods used for comparison. The visual analysis results are shown in Fig. 5. It can be seen that the clustering method proposed by us is superior to both the traditional clustering method and

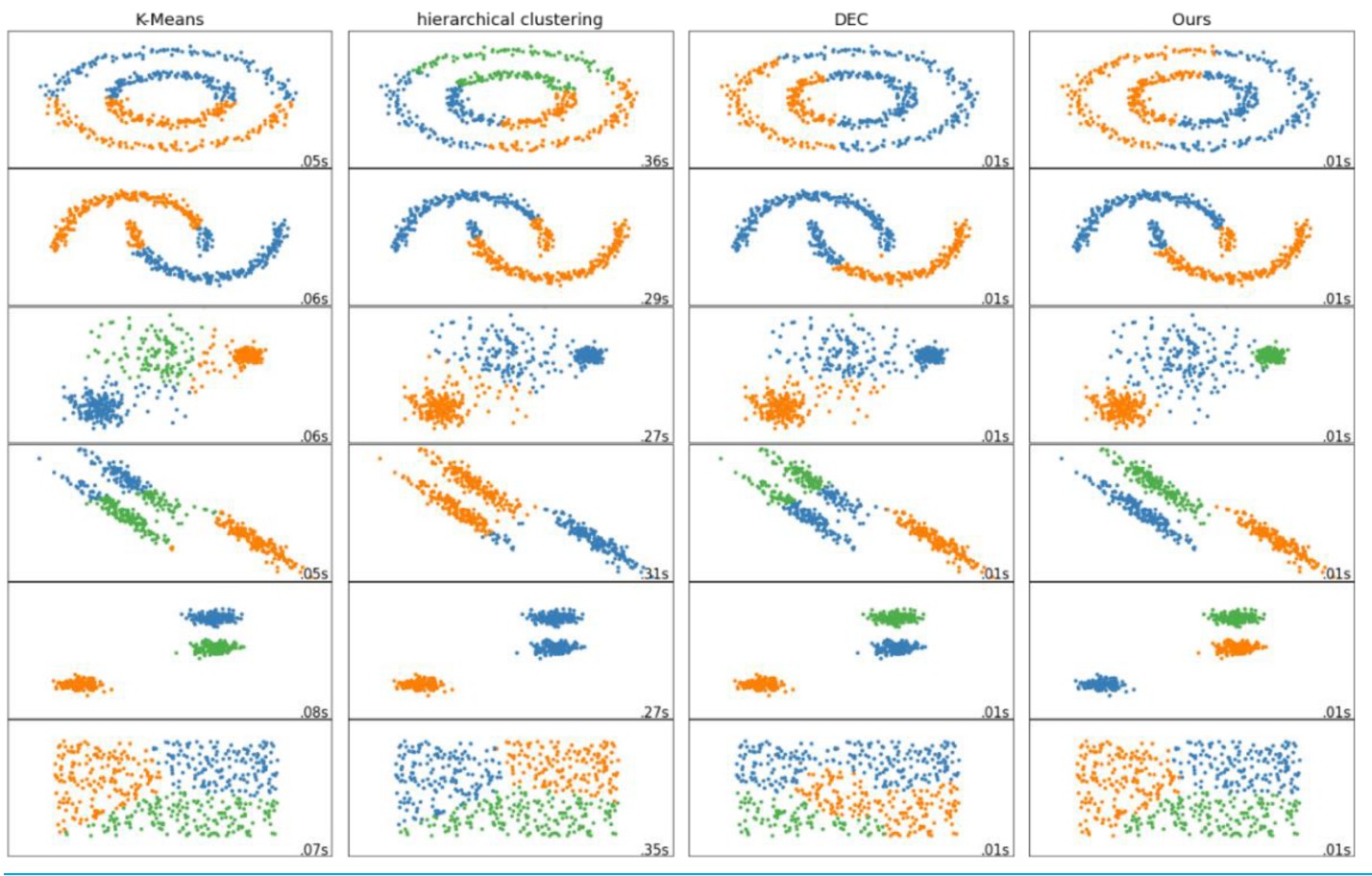

**Figure 5 Visualization comparison of different clustering methods.**

the deep clustering algorithm DEC. The clustering effect of our proposed method is better, the embedding is more obvious, the overlap is less, and the clustering of each group of nodes is better.

## Water resource liability rating results of deep clustering

In our study, we chose a water sample from Lugu Lake as the initial material for clustering analysis. The clustering results obtained are presented in Table 3. Based on the information derived from the water samples, an automated evaluation of the overall water quality of a given region can be performed, enabling the determination of its water resource liability level. The obtained results serve as evidence that the methodology proposed by our research team makes a significant contribution to the assessment of water resources. Moreover, it substantially reduces the time required for manual analysis and evaluation, offering an efficient and time-saving approach.

**Table 3 Clustering results of water resources liability levels.**

| Number | Area | Chl. a | pH | SD | DO | TN | Liability level |
|--------|------|--------|------|-----|------|-------|-----------------|
| S1 | Lianghai Sichuan Area | 0.377 | 8.25 | 6.5 | 6.56 | 0.344 | II |
| S2 | | 0.325 | 8.73 | 6.1 | 7.48 | 0.236 | |
| S3 | | 0.307 | 8.68 | 8.7 | 7.64 | 0.370 | |
| S4 | | 0.369 | 8.67 | 9.2 | 6.86 | 0.235 | |
| S5 | | 0.564 | 8.63 | 6.8 | 6.86 | 0.292 | |
| S6 | Lianghai Yunnan Area | 0.566 | 8.16 | 6.4 | 7.5 | 0.407 | III |
| S7 | | 0.754 | 8.90 | 5.5 | 7.58 | 0.524 | |
| S8 | | 0.391 | 8.33 | 5.6 | 7.06 | 0.325 | |
| S9 | | 5.184 | 8.85 | 5.7 | 8.35 | 0.434 | |
| S10 | | 0.352 | 8.67 | 5.9 | 7.62 | 0.235 | |

**Note:**
We selected Lugu Lake water sample as the initial clustering material, and the clustering results are shown in Table 3. According to the water sample information, the overall water quality of a region can be automatically evaluated to obtain its water resource liability level.

## CONCLUSION

Within the context of integrating public health data and environmental economy, the automation and intelligent assistance of water resource accounting hold immense significance. In this study, we propose an automated method for water resource liability classification in accounting, utilizing the deep clustering algorithm and the attention mechanism module. The CNN serves as the primary component of our model, harnessing its powerful feature extraction capabilities to extract intricate features and generate a feature matrix. Moreover, we enhance the CNN by incorporating an improved attention mechanism module, which introduces global information into the feature extraction process, thereby enhancing clustering accuracy while reducing the number of attention mechanism parameters. Experimental evaluations conducted on various public datasets containing complex information validate the efficacy of our proposed method. Furthermore, our specific clustering experiments on water resource debt level affirm the practicality of the proposed approach.

In future endeavors, we aim to extend the application of this deep clustering method to additional aspects of water resource accounting that involve automatic rating or classification. Additionally, we will explore the utilization of superior unsupervised methods to further enhance the clustering performance.

### Funding

This work was supported by the Research on the Influence of Block Chain Technology on the Whole Audit Process under the Background of Big Intelligence Moving Cloud (No. A2021011). The funders had no role in study design, data collection and analysis, decision to publish, or preparation of the manuscript.

## Grant Disclosures

The following grant information was disclosed by the authors:
Research on the Influence of Block Chain Technology on the Whole Audit Process under the Background of Big Intelligence Moving Cloud: A2021011.

## Competing Interests

The authors declare that they have no competing interests.

## Author Contributions

- Shiya Zhou conceived and designed the experiments, performed the experiments, analyzed the data, performed the computation work, prepared figures and/or tables, authored or reviewed drafts of the article, and approved the final draft.

## Data Availability

The code and raw data are available at Zenodo:
None. (2023). A Method of Water Resources Accounting Based on Deep Clustering and Attention Mechanism under the Background of Integration of Public Health Data and Environmental Economy [Data set]. Zenodo. https://doi.org/10.5281/zenodo.7924095.

## Supplemental Information

Supplemental information for this article can be found online at http://dx.doi.org/10.7717/peerj-cs.1571#supplemental-information.

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
