# Peer review of "A method of water resources accounting based on deep clustering and attention mechanism under the background of integration of public health data and environmental economy"

_PeerJ Computer Science, doi:10.7717/peerj-cs.1571_

## Round 0.1 · original submission · Minor Revisions

Dear Author,

Thanks for your submission to our esteemed journal, your paper has been reviewed by the experts in the field and you will see that they have lots of suggestions for improvements. I agree with them, so please revise your paper in light of those suggestions plus mine below:

1. Improve the abstract to include all the methodology details and results in the abstract.
2. Improve the language of the manuscript and remove typos/grammatical errors.

Thanks.

Reviewer 1 ·

Basic reporting

The paper discusses the importance of water resources accounting and its relation to water quality. Traditional manual methods of data analysis for water accounting are time-consuming and labor-intensive. The paper proposes an unsupervised deep clustering method using convolutional neural networks and self-attention mechanisms to address this. This method automatically classifies the debt level of water resources in different regions, aiding in overall water resource accounting. The proposed method was tested on three data sets, showing favorable results compared to other clustering techniques.
Overall, the paper looks good. However, here are a few points to improve the paper:

- Expand the literature review section to include recent and relevant research in the field. This demonstrates a thorough understanding of the existing body of knowledge and places the work in the
broader context.

- Elaborate on the methodology used, including experimental design, data collection, and evaluation metrics. Provide sufficient details to allow other researchers to replicate the experiments and validate the findings.

- Address the limitations and potential biases of the study.

- Fix the inconsistent spaces. For example, "4.2Comparison of different methods" there must be space before "Comparison."

Experimental design

Experimental designs are adequate and sufficient to support the research objectives.

Validity of the findings

Clearly present the results, including both quantitative and qualitative findings. Use appropriate statistical analysis techniques and visualizations to support the claims made in the paper.

Additional comments

Review the paper for grammar, spelling, and sentence structure errors. Ensure the writing is clear, concise, and accessible to a wide audience, avoiding unnecessary jargon or acronyms. Uncommon abbreviations should be spelled out at their first use.

Reviewer 2 ·

Basic reporting

.

Experimental design

.

Validity of the findings

.

Additional comments

By leveraging the depth clustering algorithm and incorporating an attention mechanism module, a novel automated approach for classifying water resource liabilities in accounting is proposed. The core of the approach lies in harnessing the robust feature extraction capabilities of CNN as the foundation of the model, extracting intricate details and generating a feature matrix. Moreover, an enhanced attention mechanism module is integrated within the CNN architecture, allowing for the inclusion of global information during the feature extraction process. This integration aims to enhance clustering accuracy while simultaneously reducing the number of attention mechanism parameters.
However, I have noted some concerns that need to be incorporated in the revision. Addressing these issues mentioned below would help improve the quality of the paper.

(1) The title of the article is too long and does not fully summarize the research content.

(2) The introduction should avoid repeating the information already presented in the abstract.

(3) The author should further discuss the main contributions of the research and the primary issues it addresses in the relevant field.

(4) In the literature review section, the author provides a detailed introduction to the deep clustering model. However, there is a lack of necessary logical connections among the cited references.

(5) Provide necessary explanations for Table 1 and Equation 1.

(6) After introducing Equation 4, it would be beneficial to include a summary paragraph to provide a more complete understanding of this section.

(7) Discussion section needs to be a coherent and cohesive set of arguments that take us beyond this study in particular, and help us see the relevance of what the authors have proposed.

(8) Authors need to contextualize the findings in the literature, and need to be explicit about the added value of your study towards that literature.

---

## Round 0.2 · accepted · Accept

Thank you for your improvements and now the experts are satisfied with the updated version of your article.

Reviewer 1 ·

Basic reporting

The article is updated and incorporated the comments.

Experimental design

Experimental design is sufficient to support the paper's claims.

Validity of the findings

no comment

Additional comments

The paper is improved as per the reviewer comments.

Reviewer 2 ·

Basic reporting

This paper has been well revised.

Experimental design

This paper has been well revised.

Validity of the findings

This paper has been well revised.

Additional comments

This paper has been well revised.